# Impact of Sugars on Hypothalamic Satiety Pathways and Its Contribution to Dysmetabolic States

Adriana M. Capucho [ID] and Silvia V. Conde *[ID]

NOVA Medical School, Faculdade de Ciências Médicas, Universidade NOVA de Lisboa, Rua Câmara Pestana 6, Edifício 2, piso 3, 1150-082 Lisboa, Portugal
* Correspondence: silvia.conde@nms.unl.pt

**Abstract:** Food behaviour is a complex and multifaceted cooperation between physiologic, psychological, social, and genetic factors, influencing meal timing, amount of food intake, food preferences, and food selections. Deregulation of the neurobiological mechanisms controlling food behaviour underlies the development of obesity and type 2 diabetes, two epidemics of the present century. Several brain nuclei are involved in the regulation of the different components of food behaviours; the hypothalamus is the key in controlling appetite and energy homeostasis. In this review, we will explain the role of the hypothalamus in the control of food intake and its interplay with other brain nuclei important in food behaviour. We will also highlight the deregulation of satiety pathways in type 2 diabetes and obesity and the mechanisms behind this deregulation. Finally, knowing that there are different categories of sugars and that they differently impact food behaviours, we will review in a concise manner the studies referring to the effects of sugars in satiety and reward pathways and their impacts on metabolic diseases.

**Keywords:** hypothalamus; hypercaloric diets; sugar; satiety pathways; diabetes

## 1. Introduction

In the last decades, we have witnessed an escalating number of individuals suffering from metabolic diseases, such as obesity, and its associated diseases, such as metabolic syndrome and type 2 diabetes (T2D). These metabolic diseases are genetic conditions [1] that are mainly associated with modern lifestyles; they are characterised by physical inactivity, sedentarism, and hypercaloric diets [1,2]. While obesity affects 650 million people worldwide, it is estimated that 463 million people between the ages of 20 and 79 suffer from diabetes, representing 9.3% of the world's population within this age range. In 2030, it is estimated that this number will rise to 578 million people (10.2%) and 700 million (10.9%) in 2045. What is even more worrying is that in 2019, globally, an estimated 4.2 million people died from diabetes and its complications [3]. Diabetes is a chronic metabolic condition characterised by high blood glucose levels due to the inability of the body to produce enough insulin to reduce glucose levels and due to the inefficiency of insulin, a phenomenon designated as insulin resistance [4,5]. T2D, a diabetes condition, typically begins with the development of insulin resistance. During this period, pancreatic β-cells are stimulated to increase insulin production and secreted to maintain normal blood glucose concentration. Upon diagnosis of T2D, approximately 40–50% of the β-cells are already dysfunctional and no longer able to compensate for the high levels of circulating glucose. This phenomenon results in glucose intolerance, which ultimately leads to a state of fasting hyperglycaemia [4,6]. Obesity is an established risk factor in the development of T2D, resulting from the deregulation of energy metabolism, which is also crucial in the control of T2D. The central nervous system (CNS) and, in particular, the hypothalamus, plays a crucial role in the control of energy homeostasis since it regulates functions, such as satiety and thermogenesis [7]. These functions are known to be altered in states of dysmetabolism;

therefore, they play an important role in the setting and maintenance of metabolic diseases. This manuscript focuses on and summarises the importance of the hypothalamus in the control of satiety pathways and its role in dysmetabolic states, particularly in T2D and obesity. In addition, we review the impact of sugar consumption on satiety pathways and reward systems in the development of T2D.

## 2. Control of Food Intake by the Hypothalamus

Food behaviour is a complex coordination of physiologic, psychological, social, and genetic factors influencing meal timing, the quantity of food intake, food preference, and food selection. It is regulated by several different brain circuits, where the hypothalamus is essential in controlling appetite and energy homeostasis. The hypothalamus is located below the thalamus near the pituitary gland, above the brainstem [8]. The first insight into the key role of the hypothalamus in food intake came from the observation that humans and animals with lesions in the hypothalamus brain region showed a rapid onset of obesity [9]. In fact, when the ventromedial hypothalamic nuclei (VMH) were lesioned with electrical current, the rats exhibited increased food intake and adiposity [10]. Afterward, in the 1950s, Delgado and Anand examined the effects of lesions in the VMH in cats by applying chronic (5–10 days) electrostimulation in this region; they observed an increase in food intake [11]. Of importance, the authors performed the same lesions in another hypothalamic nucleus with the same parameters, but this did not produce similar effects [11]. This nucleus is known as the "satiety center" [12]. Apart from the VMH, the lateral hypothalamic area (LHA) is also known to play a key role in the regulation of ingestive behaviour (ever since the early studies on lesions conducted by Anand and Brobeck, where they found that bilateral electrolytic lesions of LHA completely inhibited food intake to the point where the rat died of starvation) [13]. In fact, LHA is called the "feeding center" [14]. Ono et al. [15] showed later that LHA is also involved in functions associated with rewards, emotions, aversion, and learning. Although these two hypothalamic nuclei are very important in the control of food intake, the arcuate nucleus (ARC) is critical to the regulation of food intake and energy metabolism (Figure 1). The ARC has a perfect location near the median eminence (ME) that is rich in fenestrated capillaries receiving information from the blood–brain barrier. The ME plays a role in the transport of hormonal and nutritional signals from the periphery to the hypothalamus [7]. These signals are sensed by two antagonistic types of neurons: (1) the neuropeptide Y (NPY)-agouti-related peptide (AgRP)-expressing neurons (AgRP/NPY neurons), also called orexigenic neurons or appetite-stimulating neurons and; (2) the pro-opiomelanocortin (POMC) and cocaine- and amphetamine-regulated transcript (CART) neurons (POMC/CART neurons), called anorexigenic neurons/appetite-suppressing neurons. These neurons exhibit receptors for hormones, such as leptin, insulin, and glucagon-like peptide 1 (GLP-1), which regulate satiety, glucose homeostasis, insulin signalling, and energy expenditure [7,16] (Figure 1). In the POMC/CART neurons, these substances promote satiety, while in the AgRP/NPY neurons, they act antagonistically to inhibit these neurons, therefore reducing appetite and increasing energy expenditure. The POMC/CART neurons mainly project to second-order neurons in the paraventricular nucleus (PVN), but also to another hypothalamic nucleus, such as the VMH, dorsomedial hypothalamus (DMH), and the LHA. These second-order neurons have an important role in processing the received information from the ARC, projecting to other brain regions to trigger a response to maintain energy homeostasis. After food injection, POMC is cleaved to the $\alpha$-melanocyte-stimulating hormone ($\alpha$-MSH) that binds to melanocortin 3 and 4 receptor (MC3/4R) neurons in the ARC and PVN (Figure 1). Several studies reported that food intake is inhibited by hypothalamic MC4R neurons in a constant manner [17] and that MC4R activation, mainly in the PVN, leads to an increase in energy expenditure by triggering the sympathetic nervous system activation, particularly in the brown adipose tissue (BAT) [18,19]. This important role of MC4R receptors in the regulation of food behaviour and energy homeostasis is confirmed by the presence of a severe obesity phenotype in MC4R-deficient subjects [20] and by the use of MC4R agonists,

such as setmelanotide, for the treatment of genetic obesity [21]. On the other hand, in fasting conditions, ghrelin and peptide YY (PYY)—the hunger hormones secreted from the stomach and intestine, respectively—activate the AgRP/NPY neurons in the ARC nucleus, which project not only to the PVN but also to the LHA [7,22] (Figure 1). Ghrelin is secreted from the stomach mainly during starvation. The hypothalamus is the brain region that contains the highest density of ghrelin receptors, and some authors showed that ghrelin administration activates neurons in different areas of the hypothalamus, such as the ARC, VMH, and PVN [23] (Figure 1).

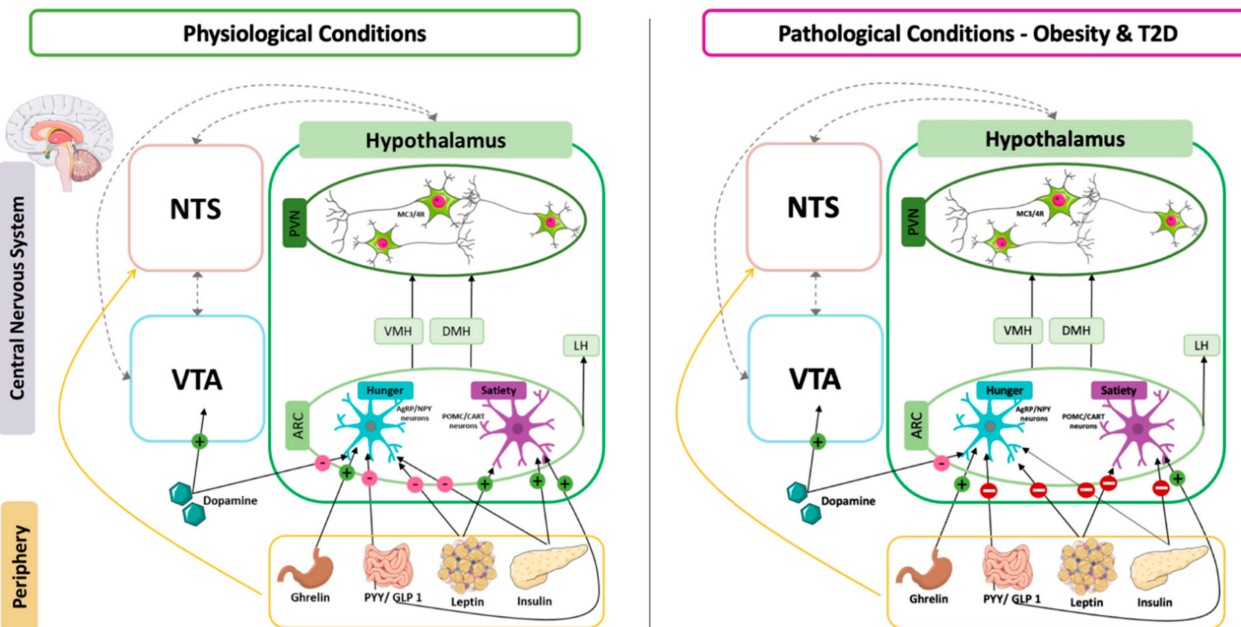

**Figure 1. Schematic representation of the brain's satiety network.** The left panel shows the neurochemistry and brain network involved in the physiological regulation of feeding behaviour and energy homeostasis and the right panel shows the same network and neurochemistry in pathological conditions, such as obesity and type 2 diabetes. Peripheral signals, such as ghrelin, PYY, GLP1, leptin, and insulin, act in certain regions of the CNS, namely the NTS, hypothalamus, and VTA. Within the hypothalamus, specifically in the ARC nucleus, the satiety-related POMC/CART neurons are activated by insulin, leptin, and GLP 1, and the hunger-related AgRP/NPY neurons are inhibited by PYY, leptin, and dopamine, and activated by ghrelin. Neuronal activity modulation of the ARC nucleus by these peripheral signals will promote alterations in other nuclei of the hypothalamus. In pathological conditions, such as obesity and type 2 diabetes, many of the peripheral signals that modulate ARC nucleus neuronal activity are impaired, with the main relevance for the blocking of insulin, leptin, and ghrelin. AgRP/NPY—neuropeptide Y-agouti-related peptide expressing neurons; ARC—arcuate nucleus; DA—dopamine; DMH—dorsal medial hypothalamus; GLP-1—glucagon-like peptide 1; LH—lateral hypothalamus; NTS—nucleus tractus solitarius; POMC/CART—pro-opiomelanocortin and cocaine- and amphetamine-regulated transcript neurons; PYY—peptide YY; VMH—ventral medial hypothalamus; VTA—ventral tegmental area.

NPY is released by the AgRP/NPY neurons to trigger the increase in food intake and reduce sympathetic activity and BAT thermogenesis, therefore modulating energy expenditure via the activating NPY type 1 receptor at the PVN, which is important in controlling the energy expenditure [24]. Additionally, AgRP can act as an inverse agonist of MC3/4 receptors preventing the anorexigenic effects of α-MSH, which may lead to a decrease in the sympathetic nervous system of the BAT and decreasing thermogenesis [7]. The AgRP/NPY neurons also have an important role in controlling POMC/CART neuronal activity since they can inhibit it via the inhibitory γ-aminobutyric acid (GABA) action. It is also important to note that both POMC/CART and AgRP/NPY neurons at the ARC

nucleus receive information from glutamatergic neurons from other hypothalamic nuclei, such as the VMH and the PVN. In contrast, the PVN receives inhibitory innervation from the LHA, which triggers the balance of feeding and regulate energy expenditure by decreasing food intake [25]. Another important brain nucleus regulating food intake, and whose information is integrated into the hypothalamus is the NTS. The NTS, in the caudal brainstem, is the first region in the brain that receives information from the alimentary tract, which is important in the control of food intake. In the NTS, a variety of nutrient, chemical, gastrointestinal–mechanical, and gut peptide signals are first integrated to control energy intake and food ingestion. The NTS is composed of different types of neuronal populations, including the ones containing elements of melanocortinergic and leptinergic signalling, crucial mediators of feeding behaviour regulation. Moreover, the ventral tegmental area (VTA) and the nucleus accumbens (NAc), two brain nuclei, are very important in controlling "hedonic" hunger, and receive projections directly from hypothalamic regions as well as from the LHA. All these brain regions constitute the mesolimbic reward system [26,27]. A lot of attention has been given recently to this mesolimbic reward system, due to its important role in the control of hedonic behaviour. The term hedonic eating refers to the intake of food driven by the reward experienced (rather than metabolic need), which is particularly relevant for cheap, highly palatable, energy-dense foods [28]. Recent studies showed that, in mice, when the inhibitory GABAergic neurons are activated in the LHA, neurons project to the VTA, and animals increase their food intake mainly due to the motivated behaviour to receive a reward [27,29]. Moreover, the activation of GABAergic neurons that project from the LHA to the VTA leads to compulsive sucrose drinking in animals, showing the importance of other circuits other than the hypothalamus in the control of food behaviour [7,27,29].

## 3. Deregulation of Satiety Pathways in Type 2 Diabetes and Obesity

Glucose is the primary energy source used by humans; it is crucial in maintaining the body's functions. The major hallmark of T2D is hyperglycaemia, which results from defects in insulin production and secretion [4,30]. This state is caused by a wavering of the regulatory circuits that maintain glucose levels within the normal range. One of the major factors that can influence the normal levels of glucose in the organism is the consumption of calorie-rich and high-fat (HF) diets. The increase in caloric intake leads to excessive adipose tissue accumulation, triggering the imbalance between what is consumed and what is spent by the organism. Several circuits and organs are required to maintain glucose homeostasis, with the brain and, particularly, the hypothalamus (the nutrient/hormone-sensing nucleus) playing central roles. In fact, the hypothalamus is not only involved in the physiological regulation of glucose homeostasis, but its deregulation (or at least the deregulation of some circuits/nuclei) contributes to the development of obesity and T2D. For example, it was reported that the ingestion of a HF diet during few days promoted an increase in the amount of saturated fatty acids (FAs) crossing the blood–brain barrier, triggering an inflammatory response in the hypothalamic neuronal population [31]. This phenomenon activates microglia and endoplasmic reticulum (ER) stress culminating in the development of central insulin and leptin resistance [32,33]. Obese individuals and some T2D patients are hyperleptinemic, due to the increased adipose tissue mass. This hyperleptinemic state results in a decrease in the ability of leptin to suppress appetite or increase the body's energy use, causing a constant increase in body weight. The inability of the body to respond to leptin is called leptin resistance [34]. Additionally, in the early stages, they are hyperinsulinemic due to the effort of maintaining normoglycemia in a state of insulin resistance [35]. In the hypothalamus, leptin and insulin resistance can result from the overactivation of their signalling cascades. In the case of leptin resistance, increased levels of leptin could lead to chronic activation of the signal transducer and the activator of transcription-3/suppressor of cytokine signalling 3 (STAT3/SOCS3), a downstream pathway activated by leptin [36]. This will, in turn, inhibit STAT3 signalling via negative feedback, resulting in leptin resistance and insulin resistance [37]. Thus, obesity and T2D

are associated with selective insulin and leptin resistance in the hypothalamus, leading to hyperphagic behaviours, alterations in glucose metabolism, and weight gain [38]. In agreement, studies performed on mice with leptin production deficiency, Lep$^{ob/ob}$ (mutations in the gene responsible for the production of leptin) and Lep$^{db/db}$ (leptin-resistance mice due to mutations in leptin receptors) [39] showed hyperphagic behaviours with impaired thermogenesis, which is associated with increased expressions of AgRP and NPY and a decreased expression of α-MSH in POMC [40]. Moreover, several studies have been performed to understand the role of brain insulin in maintaining glucose homeostasis and energy supply to the body. Several studies support that insulin might regulate NPY expression since acute insulin administration in the brain was able to reduce NPY levels in the ARC [41]. Moreover, NPY levels in the hypothalamus were shown to be increased in a rodent model of dysmetabolism, in diabetic rats injected with streptozotocin (STZ) [42]. Apart from controlling the sympathetic outflow from the PVN to the BAT and promoting an increase in thermogenesis, NPY regulates the parasympathetic outflow to the pancreas, stimulating insulin secretion [43]. These findings support the idea that insulin can control NPY levels and functions and vice-versa. Moreover, while the total lack of brain insulin receptors (IRs) leads to T2D and obese phenotypes, mice with the deletion of IRs only in the AgRP and POMC neurons do not exhibit alterations in food intake and energy homeostasis [40]. These findings clearly indicate that there are other neuronal populations apart from AgRP and POMC neurons that could be involved in the control of the central regulation of feeding by insulin, or that AgRP and POMC are not the primary integrators of insulin information-regulating feeding in the brain [41]. Nevertheless, the consumption of hypercaloric diets leading to T2D and obesity is not only associated with the impairment of leptin and insulin signalling in the hypothalamus. There is also some evidence that hypercaloric diets impair the action of ghrelin. During food deprivation, ghrelin levels increase; this is a critical signal to induce hunger during fasting. Surprisingly, and in contrast to what would be expected for an orexigenic hormone, obesity is associated, in humans and rodents, with reduced secretion and ghrelin plasma levels (for a review see [44]). In addition, in obese patients, ghrelin levels do not decrease after meals; this is consistent with the state of ghrelin resistance [44]. Several mechanisms have been postulated to justify the existence of leptin resistance in obesity, including the lower NPY/AgRP responsiveness to plasma ghrelin and the suppression of the neuroendocrine ghrelin axis [45]. This limited action of ghrelin in the hypothalamus observed in HF diet-fed animals promotes a decrease in food intake, which may be an adaptative response to prevent an increase in food intake in individuals with dysmetabolism. Nonetheless, in the PVN, ghrelin action is unaltered, which may indicate that the increase in adiposity is independent of food intake [45,46]. Although the deregulation of the satiety pathways presumes a massive impact on the development of obesity and T2D, it seems that it is not the only intervenient in the control of food behaviour and energy homeostasis, with the reward/reinforcement circuits playing a crucial role in these diseases. Recent studies demonstrated that gut hormones, such as ghrelin, PYY, and GLP-1, can modulate the response of the brain reward regions to nutrient stimuli [47].

We can conclude that homeostatic and reward circuits act together to balance eating between conditions of fasting or lack of food and conditions of overnutrition and that the disruption of these neurocircuits contributes to increased food intake, culminating in dysmetabolic states. Moreover, different types and compositions of hypercaloric diets might differentially impact these neuronal circuits regulating food behaviours. For example, increased levels of ceramides (types of saturated FAs) in the hypothalamus impair the hypothalamic control of food intake and energy expenditure [48]. Furthermore, in vitro studies performed with the hypothalamic neuronal cell line mHypoE-44 showed that prolonged exposure to palmitate attenuates insulin signalling in this hypothalamic neuronal cell line and promotes ER stress in neuronal cells, triggering lipid toxicity [49]. Additionally, in vivo studies showed that the ingestion of HF diets by animals (until there was a 10% increase in their body weights, which typically took 5–7 weeks) [50], led to an increase in hypothalamic cholesterol levels when compared to mice that were submitted to low-fat

diets, with consequent increases in food intake and body weight [48]. In another study performed in Zucker rats, an animal model of genetic obesity caused by a mutation in the gene encoding the receptor of leptin showed a link between hypothalamic ER stress caused by the increase in ceramides and the role of the nuclei in energy balance. These authors observed that hypothalamic lipid toxicity leads to a decrease in BAT sympathetic tonus and that the genetic modulation of the ceramide-induced ER pathway in the VMH increases the sympathetic nervous system-mediated BAT thermogenesis, as well as insulin signalling, promoting an overall improvement in the metabolism of the Zucker rats [48,51]. Nevertheless, when talking about the impact of hypercaloric diets on food behaviour, especially in humans, one cannot forget that hypercaloric diets include high percentages of sugars in their composition. In fact, research shows that sugar intake differentially affects the anorexigenic/orexigenic pathways, with sucrose consumption in mice leading to a temporary decrease in orexigenic peptides followed by activation of the orexigenic pathway, potentiating caloric consumption [52]. Moreover, besides the percentage of sugar in the diet, the types of sugars present in the diet may differently impact food behaviour.

## 4. Impact of Sugar Consumption on Food Behaviour

Sugars are carbohydrates that include fructose and glucose (monosaccharides), and lactose and sucrose (disaccharides), playing different roles in the organism. Sugars can be categorised as (1) intrinsic/natural and (2) extrinsic/added, depending on if they are present in the food without processing, or if they are added. Sucrose, fructose, glucose, starch hydrolysates, and other isolated sugar preparations added during food preparation and manufacturing are included in the added sugars category [53]. It seems consensual that added sugars have a noxious role in the development of metabolic diseases and that sugars that are intrinsically present in food seem to have more harmless impacts on dysmetabolic conditions [54,55]. For example, in a study performed by Monteiro-Alfredo [55], it was shown, in Goto-Kakizaki rats, that the ad libitum ingestion of sugary solutions for 4 weeks impaired energy balance regulation, leading to higher caloric intake, weight gain, fasting hyperglycaemia, insulin intolerance, and impaired oxidative stress/glycation markers than the ad libitum intake of fruit juices, demonstrating the different impacts of added vs. intrinsically present sugars in the metabolism.

Importantly, we cannot oversimplify the impact of the different added sugars as it seems that they differently affect satiety pathways. For example, it was shown that the intracerebroventricular administration of glucose and fructose have contrary effects on food intake, with glucose suppressing food intake via the inactivation of hypothalamic AMP-kinase causing the activation of malonyl–CoA signalling system and fructose having inverse effects and, thereby, increasing food intake [56]. These results agree with the data found in a more recent study performed on rats, where the effects of 24 h of free access to different sugars, e.g., sucrose, glucose, fructose, or high-fructose corn syrup on hypothalamic appetite regulation were accessed. The authors observed that glucose consumption resulted in the upregulation of seven satiety-related hypothalamic peptides, including cholecystokinin (CCK), whereas fructose decreased CCK, suggesting that glucose might have a greater impact in promoting satiety when compared to fructose [57]. Interestingly, high fructose corn syrup, a sweetener commonly used to enhance the flavour of foods and beverages, as well as sucrose, had no effect on hypothalamic CCK [57], suggesting that the deregulation of other neural mediators might be involved in the deregulation of hypothalamic pathways by these sugars. Another thing that should be taken into consideration is the fact that these studies were performed in response to acute administrated sugars and that the probability of the long-term intake of sugars might have different effects on hypothalamic satiety pathways. In a study dedicated to evaluating the effects of sucrose, glucose, and fructose in peripheral and central signals, the authors tested their effects after 24 h, 1 week, and 2 weeks of administration in rats and found that long-term exposure to the different sugars differently impacted satiety pathways [58]. Moreover, they found that a 2-week intake of sugar solutions resulted in the downregulation of hypothalamic NPY

mRNA, in which a sucrose or fructose solution leads to the upregulation of hypothalamic CB1 mRNA, and that glucose or fructose downregulated hypothalamic POMC mRNA [58]. In accordance with these results and with the role of endocannabinoids in the regulation of sugar intake, the ingestion of fructose for 1 week was found to affect enzymes involved in the synthesis and degradation of hypothalamic endocannabinoids [59]. Nevertheless, these results might not be so easily translatable to humans. Some authors have found that sugar intake, fructose and glucose, modified serum PYY without changing plasma leptin and ghrelin levels [60], which contrasts with the data found in rats [58]. Moreover, they found that—by using magnetic resonance imaging—glucose, but not fructose, was able to quickly (within 15 min) mediate satiety by reducing brain activity in the hypothalamus [60]. Of interest, they also found that glucose intake induced an increase in functional connectivity between the hypothalamus and striatum, suggesting that glucose improves the communication between appetite control centres.

Apart from impacting hypothalamic satiety pathways, sugar also impacts reward systems due to its additive, palatable, and rewarding characteristics, which may lead to compulsive eating [30,61]. As described in Section 2, hypothalamic neurocircuits project to the mesolimbic pathway, which is composed of the VTA and the NAc, called the reward system, and is important in the control of hedonic hunger [7,16]. Dopamine (DA) is an important neurotransmitter that is involved in functions, such as cognition, emotive behaviour, reward, and memory, also playing a critical role in the control of feeding behaviour and food intake [62]. It is a master regulator of food intake via the mesolimbic neurocircuit by modulating the motivational processes associated with appetite, through its projections from the VTA into the NAc and from the NAc to the hypothalamus [62]. In humans, the ingestion of palatable food promotes the release of DA in the dorsal striatum in proportion to the self-reported level of pleasure derived from eating food [63]. Moreover, upon first exposure to a food reward, DA neurons in VTA increased their firing, resulting in an increase in DA release in NAc [64]. However, the involvement of DA in the reward is more complex than the mere encoding of the hedonic value. Some reports have shown that metabolic mediators from the periphery, such as leptin, insulin, and ghrelin interact with DA in the brain [22]. The increased consumption of foods enriched in sugars can impair the homeostatic mechanisms that control eating behaviour, as in the cases of the dopaminergic pathways. This may lead to overweight, T2D, and obesity states. Previous studies performed on animals and humans showed that glucose as well as sucrose [65] modulate DA activity in the VTA and substantia nigra. More recently, it was shown that post-ingestive sucrose, but not sucralose—an artificial sweetener and sugar substitute—can sustain operant food-seeking behaviour, which is an effect be mediated the activation of a subpopulation of VTA dopamine neurons via the vagus nerve [66]. Sucrose is broken down into fructose and glucose molecules and, therefore, fructose may possess reinforcing properties activating the reward system, such as sucrose. In fact, it has been shown that fructose activates reward-related regions within the mesocorticolimbic DA system, as seen by c-Fos induction in the dorsal striatum and amygdala in rats, increases the BOLD signal in the dorsal striatum, pre-frontal cortex, and orbitofrontal cortex of pigs, and increases dopamine levels in the VTA (for a review, see [67]). Moreover, it was shown that fructose-mediated cortical disinhibition increases impulsivity toward food rewards, potentiating overfeeding [67]. Overall, we can conclude that different sugars differently regulate hunger and satiety but also food-seeking and reward pathways.

Furthermore, the effects of sugars on the modulation of dopamine pathways in the mesolimbic system might not be direct and could involve the release of hormones in the periphery and the interaction with the dopaminergic pathways [68]. Insulin, ghrelin, leptin, and GLP-1 interact with DA neurons in the midbrain [68–71]. Leptin and insulin inhibit DA neurons while ghrelin triggers its activation. Insulin is known to promote the desire for fat and sugars, reducing hedonic feeding. Moreover, upon injections of leptin in the VTA, food intake is reduced, and the ablation of leptin receptors in this region increases the reward of palatable meals, which include sugars. In the presence of dysmetabolism

states, the modulation of dopaminergic pathways with DA agonists triggers weight loss in obese animals through the activation of dopamine type 1 (D1R) and type 2 receptors (D2R), although D2R is more associated with food seeking, motivation, and satiety inhibitory control [72]. Additionally, hyperphagia in obese mice is attenuated with DA administration and reward.

## 5. Conclusions

In conclusion, we provide evidence that not all sugars are equally deleterious in affecting the control of food behaviours, as they have different impacts on the deregulation of satiety and reward pathways leading to obesity and T2D. Knowing that food behaviours involve complex and multifaceted interplays of mechanisms, driven not only by hunger/satiety but also by cravings and hedonic memories, more attention and research should be performed to study the impacts of the different sugars in these pathways, which are particularly important in states of dysmetabolism, such as obesity and T2D.

**Author Contributions:** Conceptualization, A.M.C. and S.V.C.; writing—review and editing, A.M.C. and S.V.C. All authors have read and agreed to the published version of the manuscript.

**Funding:** This study was supported by the Portuguese Foundation for Science with a research grant EXPL/MED-NEU/0733/2021.

**Institutional Review Board Statement:** Not applicable.

**Informed Consent Statement:** Not applicable.

**Data Availability Statement:** The data presented in this study are available on request from the corresponding author.

**Conflicts of Interest:** The authors declare no conflict of interest.

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
