# Peer review of "Impact of Sugars on Hypothalamic Satiety Pathways and Its Contribution to Dysmetabolic States"

_diabetology, doi:10.3390/diabetology4010001_

Round 1

Reviewer 1 Report

The manuscript entitled "Impact of sugars on hypothalamic satiety pathways and its contribution to dysmetabolic states" in which the authors focused and summarized the importance of the hypothalamus in the control of satiety pathways and its role in dysmetabolic states, especially in type 2 diabetes (T2D) and obesity. They also demonstrated the impact of sugars consumption in satiety pathways and reward systems in the development of T2D.

The work is understandable and the topic is important. The scientific narrative is well structured and flows naturally from one idea to the next.

However, this paper suffers from few shortcomings that if modified would make the manuscript very suitable for publication in Diabetology Journal.

Shortcomings:

1-    The authors write “Interestingly, and in contrast, with the hyperphagic phenotype of obese and T2D people, rats under HF diet exhibited ghrelin resistance by reducing NPY/AgRP responsiveness to plasma ghrelin and suppressing the neuroendocrine ghrelin axis to limit further food intake [44]”.  Is there difference of ghrelin resistance between the hyperphagic phenotype of obese and T2D people and the rats with HF diet? Please clarify.

2-    Please define (GLP-1”, which ……………………..[7]) in the first mention then write the abbreviation after that.

3-    Please modify activation of activator of transcription 3/suppressor ….. (STAT3/SOCS3 to “activation of Signal transducer and activator of transcription-3/ ……………”.

4-    The scientific narrative is well structured and understandable. However, the language and grammar used in the whole manuscript need significant improvement. Please revise the whole paper for the language and grammar.

Below is some advice to change (related to typos and language):

·       “hedonic” huger receive projections directly from hypothalamic…….….. system [26],[27]. Please correct huger to “hunger”.

·       “Recent studies showed in mice that when the inhibitory GABAergic neurons are activated in the LHA neurons that project to the VTA, animals increase their food intake mainly due to a motivated behaviour to receive a reward [27]”.

Please add ,

·       Also, the activation of GABAergic neurons that project from the LHA to the VTA lead to……………. [27],[29],[7]. Please correct lead to “leads”.

·       Also, NPY …………dysmetabolism - the diabetic rats injected with streptozotocin (STZ) [42].

Please add dysmetabolism “in the” diabetic rats injected with streptozotocin (STZ).

·       Please rephrase “Apart from controlling the sympathetic outflow from the PVN to the BAT therefore increasing thermogenesis, NPY acts to control parasympathetic outflow to the pancreas, stimulating insulin secretion [43]”.

·       Please rephrase “Also, in vitro studies performed with the hypothalamic neuronal cell line mHypoE-44 showed that prolonged exposure to palmitate leads to endoplasmatic reticulum (ER) stress triggering hypothalamic lipid toxicity with impact on insulin signaling, assessed as semiquantitative one-step RT-PCR [48 ]”.

Author Response

We acknowledge the reviewer the comments made to our manuscript. We have answered to the shortcomings raised and hope that in the present form the manuscript is suitable to publish in Diabetology Journal.

The shortcomings raised by the reviewer are in bold and the answer in italic style.

Shortcomings:

1-    The authors write “Interestingly, and in contrast, with the hyperphagic phenotype of obese and T2D people, rats under HF diet exhibited ghrelin resistance by reducing NPY/AgRP responsiveness to plasma ghrelin and suppressing the neuroendocrine ghrelin axis to limit further food intake [44]”.  Is there difference of ghrelin resistance between the hyperphagic phenotype of obese and T2D people and the rats with HF diet? Please clarify. The reviewer highlighted a good point, as the concept of ghrelin resistance and if there was or not differences between humans and rodents, were not clear. We have rephrased and now it can be read: “During food deprivation ghrelin levels increase this being the critical signal to induce hunger during fasting.  Surprisingly, and in contrast to what it would be expected for an orexigenic hormone, obesity is associated in humans and rodents with a reduced secretion and reduced plasma levels of ghrelin (for a review see  [44]). In addition, in obese patients ghrelin levels do not decrease after meals this being consistent with a state of ghrelin resistance [44]. Several mechanims have been postulated to justify the existance of leptin resistance in obesity, including  the lower NPY/AgRP responsiveness to plasma ghrelin and the suppression of the neuroendocrine ghrelin axis  [45]. This limited action of ghrelin in the hypothalamus observed in HF diet fed animals, promotes a decrease in food intake, which may be an adaptative response to prevent an increase in food intake in individuals with dysmetabolism.”. Hopefully this topic is now clearer.

2-    Please define (“GLP-1”, which ……………………..[7]) in the first mention then write the abbreviation after that.  We have included the definition of GLP-1 in figure legend as well as in the first mention to GLP-1 in the text.

3-    Please modify activation of activator of transcription 3/suppressor ….. (STAT3/SOCS3 to “activation of Signal transducer and activator of transcription-3/ ……………”. We modified the definition of STAT3/SOCS3.

4-    The scientific narrative is well structured and understandable. However, the language and grammar used in the whole manuscript need significant improvement. Please revise the whole paper for the language and grammar. We revised the manuscript for language and grammar use. We corrected all the typos and language pointed by the reviewer as well as others that we found throughout the text.

Reviewer 2 Report

I thank the journal for giving me an opportunity to review the article titled “Impact of sugars on hypothalamic satiety pathways and its contribution to dysmetabolic states” by Capucho and Conde. The review is overall well-written and detailed, describing the role and connection between the hypothalamus, dysregulated satiety pathways in T2D and obesity and the mechanisms related to this dysfunction. There are some minor changes that the authors are requested to make.

1.       Figure 1: Include panel “A” (left) and “B” (right) in the actual figure for clarity and mention these in the legend as well.

2.       Use abbreviations consistently throughout manuscript. For example – in conclusions use “T2D” abbreviation

3.       Figures to summarize sections 3 (deregulation of satiety pathways in type 2 diabetes and obesity) and 4 (Impact of sugar consumption in food behavior) would be helpful for the readers.

4.       Minor language and grammar edits needed throughout manuscript. For example – (i) Introduction section: Obesity is “an established”. (ii) Rather than using collective nouns (“we”) at several places, the authors are requested to write sentences in third person. (iii) Language correction – conclusion section: “However, and knowing…” (incorrect). 

Author Response

We acknowledge the reviewer the comments made to our manuscript. We hope to have addressed all the items requested. 

The points raised by the reviewer are in bold and our answer in italic style.

  1. Figure 1: Include panel “A” (left) and “B” (right) in the actual figure for clarity and mention these in the legend as well. In order to not interfere with the design of the figure we decided to designate figures as left panel and right panel. We hope that is suitable for a better understanding.
  2. Use abbreviations consistently throughout manuscript. For example – in conclusions use “T2D” abbreviation. We revised the abbreviations and now we believe that we are using abbreviations consistently throughout the manuscript.
  3. Figures to summarize sections 3 (deregulation of satiety pathways in type 2 diabetes and obesity) and 4 (Impact of sugar consumption in food behavior) would be helpful for the readers. This review was a commissioned review, and the guidelines were that we only could include a figure in the manuscript. Figure 1 aims to summarize sections 2 and 3 of the manuscript, with left panel summarizing section 2 and right panel summarizing section 3 (deregulation of satiety pathways in type 2 diabetes and obesity).
  4. Minor language and grammar edits needed throughout manuscript. For example – (i) Introduction section: Obesity is “an established”. (ii) Rather than using collective nouns (“we”) at several places, the authors are requested to write sentences in third person. (iii) Language correction – conclusion section: “However, and knowing…” (incorrect). We acknowledge the reviewer having highlighted these incorrections. We have revised the typos and the grammar throughout the manuscript.